# Qualitative Investigation into the Mental Health of Healthcare Workers in Japan during the COVID-19 Pandemic

**DOI:** 10.3390/ijerph19010568

**Published:** 2022-01-05

**Authors:** Yasuhiro Kotera, Akihiko Ozaki, Hirotomo Miyatake, Chie Tsunetoshi, Yoshitaka Nishikawa, Makoto Kosaka, Tetsuya Tanimoto

**Affiliations:** 1School of Health Sciences, University of Nottingham, Nottingham NG7 2HA, UK; 2Department of Breast Surgery, Jyoban Hospital of Tokiwa Foundation, Iwaki 972-8322, Japan; aozaki-tky@umin.ac.jp; 3Medical Governance Research Institute, Tokyo 108-0074, Japan; tetanimot@yahoo.co.jp; 4Orange Home-Care Clinic, Fukui 910-0018, Japan; hmiyatake@orangeclinic.jp (H.M.); m.kosaka0811@gmail.com (M.K.); 5Department of Community Health Nursing, University of Fukui, Fukui 910-1104, Japan; sk21803@g.u-fukui.ac.jp; 6Department of Health Informatics, Kyoto University School of Public Health, Kyoto 606-8501, Japan; yoshitakanishikawa@gmail.com

**Keywords:** healthcare workers, Japan, mental health, COVID-19, coping, intrinsic rewards, self-care

## Abstract

The COVID-19 pandemic has negatively impacted the mental health of healthcare workers in many countries including Japan. While many survey-based findings have reported the serious state of their wellbeing among healthcare workers, the first-hand experience of the mental health and coping in this population remains to be evaluated. Accordingly, this study aimed to appraise them using constructionist thematic analysis on semi-structured interviews attended by a purposive and snowball sample of 24 healthcare workers in Japan conducted in December 2020–January 2021. Four themes were identified: (1) increased stress and loneliness, (2) reduced coping strategies, (3) communication and acknowledgement as a mental health resource, and (4) understanding of self-care. Participants noted that the characteristics of Japanese work culture such as long hours, collectivism and *hatarakigai* (i.e., meaning in work) to explain these themes. These findings suggest that robust support at an organizational and individual level, capturing intrinsic values, are particularly important for this key workforce to cope with increased stress and loneliness, leading to better patient care.

## 1. Introduction

The mental health of healthcare workers has been negatively impacted by the COVID-19 pandemic [1]. For example, in the United Kingdom, the rates of depression, anxiety and stress among healthcare workers have quadrupled from pre-COVID-19 to after the first wave, April–May 2020: the pre-COVID prevalence of severe depression (5%), anxiety (8%) and stress (11%) raised to 21%, 36% and 46%, respectively, among healthcare workers [2]. In the United States, nearly half of healthcare workers experienced serious mental health symptoms including suicidal ideation [3]. Likewise, in Japan, about one-third of healthcare workers experienced burnout [4]. Commonly, fear of infection, close contact with COVID-19 patients, lack of personal protective equipment and lack of information/guidance were noted as the primary risk factors for their mental health [5,6]. Indeed, healthcare workers are regarded as essential workers, treating COVID patients, risking their own lives: they are seven times more likely to be infected than other occupational groups [7]. Especially, during the initial phase of the COVID-19 pandemic, healthcare workers were forced to work in different contexts and roles without sufficient information and guidelines. They must make practical, sometimes seemingly ‘inhumane’ decisions to prioritize care with limited medical resources. As illustrated in the Job Demand-Control-Support model—one established workplace wellbeing model describing how job demands can cause stress whilst job control and support can help cope with the stress [8]—, these increases in job demands and decreases in job control and support can add extra burden to their wellbeing [9]. Moreover, inequality of care access has become more salient in the pandemic, adding extra stress to healthcare workers [10]. However, research thus far primarily has focused on the quantitative findings, missing the first-hand experience of healthcare workers serving the public during the COVID-19 pandemic. Hence there is a need to appraise the lived mental health experiences of this worker group. Coping strategies, communication and self-care were focused on in this study due to their relevance to workplace mental health. Coping was identified as a key factor to maintain a high level of mental wellbeing among healthcare workers during the pandemic [11,12]. Likewise, workplace communication can also play a crucial role in employee mental health [13,14]. Lastly, the importance of self-care has been increasingly highlighted especially in healthcare workers [15,16,17].

### 1.1. Coping Strategies

To maintain a high level of mental wellbeing, establishing effective coping strategies is essential. Previous workplace studies reported that knowledge in the variety of coping strategies would help maintain their mental health [18]. In general, positive coping strategies—such as help-seeking, meditation, counseling and humor—are conducive to wellbeing and quality of working life, whereas negative ones—such as avoidance, substance abuse, self-harm—are risk factors for poorer wellbeing and quality of working life [11,19]. During the COVID-19 pandemic, several effective coping strategies were reported among nurses in China such as team communication and pro-social behaviors [20]. While these findings offer helpful insights about coping among healthcare workers, how COVID-19 impacted healthcare workers’ existing coping strategies remains to be appraised.

### 1.2. Communication in Workplace

In the restricted working settings, how colleagues communicate with each other has changed. Safe and clear workplace communication was noted as an effective coping strategy [20]. This was highlighted during the pandemic, where healthcare workers faced many uncertain situations. For example, among Japanese nurses, an establishment of standard protocol was a key mental health resource that made them feel safe, enabling more compassionate care for patients [21]. Indeed, communication has been noted as a key factor for good workplace mental health. In an organization where employees feel that they communicate with each other well, the level of mental wellbeing tends to be high [22]. Communication among colleagues had a great positive impact on employees’ mental wellbeing [23]. Workplace communication can facilitate authentic dialogues among colleagues, which are associated with higher wellbeing [24]. In the current ‘second pandemic’, that is a mental health crisis after the COVID-19 pandemic, the importance of workplace communication has been highlighted ever before [25], therefore this study evaluated how that relates to healthcare workers’ mental health.

### 1.3. Emerging Importance of Self-Care

Self-care has been increasingly attracting attention in recent mental health studies, especially among healthcare workers. This emphasis has been further accentuated during the pandemic, ensuring healthcare workers to maintain a high level of mental health [26]. Indeed, many healthcare workers, who have been trained to care for others, feel guilty caring for themselves, underestimating the importance of self-care [15]. However, self-care is crucial for healthcare workers to cope with the increasing workload and uncertainty, and to balance their work life and the other areas of life based on their own cultural backgrounds. Unsurprisingly more healthcare professionals today acknowledge the importance of self-care: 100% of Australian healthcare workers who practice self-care reported it is effective, and 70% of those who do not practice wished to do so [27]. Reviews on the wellbeing of healthcare workers during the COVID-19 pandemic in western countries identified self-care as a professional imperative [28,29], and evidence-based strategies were suggested including spiritual practices, relaxation, focusing on important relationships (e.g., family), healthy sleeping and diet, and meditation [30]. However, how Japanese healthcare workers view and practice self-care during the COVID-19 pandemic remains to be examined. 

### 1.4. Study Aims

This study aimed to appraise the first-hand experience of mental health among healthcare workers in Japan. Using semi-structured interviews, we explored specific challenges they have encountered during the pandemic, what they have found helpful to cope with the mental health difficulties, and what may be effective to support the mental health of this population group in the future. Three research questions were established, exploring the impact of COVID-19 on the mental health of healthcare workers in Japan (RQ1), their coping (RQ2), and their thoughts on the improvement of mental health (RQ3). 

## 2. Materials and Methods

### 2.1. Research Design

Thematic analysis of in-depth semi-structured interviews attended by 24 healthcare workers in Japan (14 males and 10 females; Age M = 34.29, SD = 6.45, Range 27–49 years old; 14 doctors, 2 nurses, 6 physiotherapists, and 2 administrators; Table 1) was performed within a social constructionist framework, evaluating how their experiences are created based on the data [31]. The eligibility criteria for participation were 18 years or older, and a healthcare worker who was working or had worked during the time of restrictions due to COVID-19. This study adhered to the consolidated criteria for reporting qualitative research [32].

### 2.2. Procedure and Analysis 

The university research ethics committee approved this study (No. ETH2021-0101). Participants were recruited through purposive and snowball sampling methods: an initial announcement about the study was disseminated through authors’ professional networks. The healthcare workers who responded to the announcement were invited to the interview with the study information and consent form.

We used the semi-structured interview method to collect detailed information and allow participants to express their experiences, feelings and thoughts [33]. This interview method is particularly advantageous for appraising complex issues, because it allows researchers to add follow-up questions to cover information that could be missed in a standardized data collection method such as close-ended surveys [34]. As the mental health of healthcare workers in Japan during COVID-19 pandemic has not been evaluated in depth, we employed this method to counter the possibility of missing information.

Prior to the interview, all participants received the pre-designed interview questions, which focused on mental health and coping (Appendix B). Due to the physical distance restrictions, all interviews were conducted online. The interviews, conducted in December 2020 and January 2021, were recorded and transcribed verbatim, which were confirmed for accuracy by the participants after the interview. The study time fell between the second and third waves of infections in Japan, which was before the rollout of vaccination in February and overlapped with the Second State of Emergency, 8 January to 21 March 2021. 

Thematic analysis was used to analyse the interview data. This analysis method helps to organize and identify patterns of meaning (i.e., themes) throughout a dataset in a systematic manner [35]. Thematic analysis is particularly helpful to identify meanings and understand idiosyncratic experiences. Moreover, it is noteworthy that the main function of thematic analysis is not related to commonality in data, rather it evaluates the importance and relevance to the research questions to identify themes. The six steps suggested by Braun and Clarke [36] were followed. 

The lead author Y.K., an accredited psychotherapist and mental health researcher, interviewed all participants and transcribed the interview data. Y.K. then analysed the data using thematic analysis. The other co-authors helped with recruitment and reviewed the analysis. In order to retain coherence and transparency of analysis, an investigator triangle was formed with a psychology researcher and health researcher [37]. All themes have been checked and agreed upon by all co-authors and researchers, and later by all participants.

#### 2.2.1. Familiarization

Interview data was read repeatedly to understand the entire data and gain initial interpretations and patterns, informing possible themes [35,36,38]. 

#### 2.2.2. Generating Initial Codes

To start the systematic analysis of data, coding was conducted offering labels to data [35]. The theory-driven approach [36] for coding was utilized, relating to our research questions.

RQ1: What was the mental health impact of COVID-19 among healthcare workers in Japan?

RQ2: How did they cope with the impact?

RQ3: How can their mental health be improved?

One hundred and five codes were identified (Appendix C) including stress, loneliness, COVID stigma, affable atmosphere, self-care, social gathering, and acknowledgement. A data corpus and mind-mapping were used for transparency and coherence [39]. 

#### 2.2.3. Searching for Themes 

The codes were organized into potential themes. We used the mind-map method to see all the codes at the same time, and move and connect them freely [36]. The 105 codes were grouped together into four themes: increased stress and loneliness (T1), reduced strategies for coping (T2), communication and acknowledgement as a mental health resource (T3), and understanding of self-care (T4).

#### 2.2.4. Reviewing Themes

Next, those four themes were reviewed to see whether the themes accurately capture the relevant dataset [35]. Codes were compared with the relevant data extracts (see the Results section) to ensure the coherence between each theme and each set of extracts [36] more specifically (a) themes capture the most important elements of the data and (b) themes are relevant to the research questions [35].

The data were organized to address our research questions. The increased levels of stress and loneliness that healthcare workers experienced during the pandemic (T1) corresponded to RQ1; reduced coping strategies for mental health difficulties (T2) corresponded to RQ1; supportive communication in the workplace, and acknowledgement of their work helped to protect their mental health (T3), which answered RQ2; and more understanding and positive regard to self-care as an effective mental health approach (T4) addressed RQ3. 

#### 2.2.5. Defining and Naming Themes

The essence and range of the collated data were reviewed to establish that each theme is presented to accurately demonstrate the accompanying narrative [35].

## 3. Results

The datasets from T1 ‘increased stress and loneliness’ considers that healthcare workers experienced more stress and loneliness during the pandemic than before. There were no established guidelines and experts to consult; healthcare workers were faced with uncertainty, while treating ever-increasing COVID-19 patients. Moreover, social expectation that healthcare workers cannot be infected with COVID-19, forced them to more limited daily life activities and more alert precautions, leading to a sense of loneliness. Loneliness was also experienced in the workplace, with segmented rooms and limited interactions. These lead to T2 ‘reduced strategies for coping’ that entails no informal gatherings among colleagues, which have been an effective way to connect with colleagues and create a supportive workplace culture. Additionally, other personal activities such as visiting family members and going shopping were restricted, making it harder to support their own mental health. However, while recognizing these barriers, healthcare workers identified T3 ‘communication and acknowledgement as a mental health resource’ reporting that these were particularly helpful to maintain their mental health. Lastly, healthcare workers suggested T4 ‘understanding of self-care’ in their work culture can contribute to improvement of mental health in this sector. Findings are summarized in Table 2.

### 3.1. Theme 1: Increased Stress and Loneliness

All participants reported that since the outbreak, their stress level and sense of loneliness have increased, associated with various factors including uncertainty, increased work demands, fear of being infected, and social pressure for healthcare workers.


*P1 (Participant 1): There were no established guidelines nor legal support and protection for us. I had to look up many things and interpret various pieces of information. … I also had to report our cases to the management. These have significantly increased my workload.*



*P16: As a healthcare worker, I must not be infected, as that will impact the hospital’s reputation. This means that my family also had to restrict their daily behaviours, which was especially stressful for my children.*


Additionally, some commented on the stigma related to COVID-19, along with a discrimination and harsh criticism for those who are infected. 


*P10: There is a discrimination against people who are infected. They are unreasonably criticized [for being infected]. Anyone could be infected. Sometimes it’s not within their control. Staff at a care home, where a cluster was found, was and still is criticized.*



*P21: If people know that there is a positive case among staff at the hospital, it will be a big deal. Everyone at the hospital thinks ‘I don’t want to be the first one’. Stigma for COVID is strong. We had a patient who was infected, but she moved to another town because people in the community were harsh to her.*


These comments suggest that negative views placed onto people who are infected, especially healthcare workers, added another layer of stress and loneliness. 

### 3.2. Theme 2: Reduced Strategies for Coping

Participants reported that their limited workplace and daily life activities exacerbated their mental health. Some of the healthful activities, which they used to engage with pre-COVID, were prohibited during the pandemic, compromising their ability to cope with mental distress. 


*P22: We used to have a lot of chitchats, for example, at the end of our shifts. While writing a daily report, we also talk about how our families are or what we did on a weekend. … During a shift, sometimes we must have direct, negative or intense conversations, but chitchats will help retain our relationship: you know that the person doesn’t dislike you.*



*P19: As a physiotherapist, what I can do for my patients is now limited as my work usually involves direct touch on the patient’s body. I feel less of the meaning of work, hatarakigai, as now I don’t feel like I am a physiotherapist sometimes.*


In addition to their workplace activities, their activities outside the workplace, which could help their mental health, are also limited.


*P13: We cannot travel, meet with friends, and engage with hobbies, so there is no way to destress ourselves. Moreover, we cannot have social gatherings with colleagues, which now I realize, are very important for our wellbeing, knowing each other better.*


Restrictions associated with the pandemic reduced the range of healthful activities inside and outside the workplaces, making it hard for healthcare workers to cope with increased distress.

### 3.3. Theme 3: Communication and Acknowledgement as a Mental Health Resource

Despite the limited strategies for coping, healthcare workers identified some mental health resources that help them counter mental distress, including supportive workplace communication, where they feel safe to discuss difficult issues [40], and acknowledgement of their hard work from their colleagues or patients.


*P3: It is very helpful to connect with healthcare workers who are in a similar circumstance to me. … This kind of conversation happens organically in the face-to-face context, but now we need a video call to do that.*



*P10: I feel comfortable talking about my mental distress with my colleagues.… Everyone is available and willing to help if I need to talk.*


These comments indicate the importance of communication to their mental wellbeing. Participants recognized the importance of communication even more during the pandemic. Moreover, acknowledgement of their work was also noted as a protective factor for mental health. 


*P5: Positive feedback from my line manager or the head of the hospital helps me cope with stress. Also, some patients brought me some gifts, appreciating my treatment. That kind of moment is helpful for my mental health.*



*P20: Today many healthcare workers who work with COVID patients are featured in a TV programme or a section in news shows. Because of that, people’s understanding towards those workers has been increasing. But those who don’t directly work with COVID patients are also impacted. They want to be acknowledged too. In many cases, just a ‘thank you’ would be enough.*


Healthcare workers reported that acknowledgement of their work from their colleagues and patients helped or would help to cope with occupational stress and protect their mental health.

### 3.4. Theme 4: Understanding of Self-Care

To improve the mental health of healthcare workers, participants suggested that it is essential to have an increasing understanding of self-care. They believe that a good workplace understanding of self-care may be helpful for Japanese healthcare workers to achieve a high level of mental health.


*P4: My line manager believes that if we don’t take good care of ourselves, we cannot take care of others. He supports self-care, which positively impacts our workplace culture. I am very thankful to him for that.*



*P7: My team endorses self-care; it may be because we are in palliative care. If I think about the culture among doctors in Japan, I don’t think that’s the case.*


At the same time, under-emphasis and difficulty of self-care in the current practice was also noted. Often this was discussed in relation to Japanese work culture. 


*P16: I think it’s in Japanese culture. We cannot say we are suffering, or we are in pain, because other people may be also … Japanese people are not good at taking care of themselves. That is a taboo, you cannot say that in this culture.*



*P8: As a doctor, I find it hard to self-care. … The root of this is a value of Japanese people during WWII, ‘We don’t ask for anything until we win’. We believe that asking for something means we are not cooperating, but in reality, we need to care for ourselves, before care for others. … It’s been a challenge for me to take good care of myself.*


Participants highlighted those aspects of Japanese culture, especially among older generations, which may hinder their attitudes towards taking care of themselves. However, they are also aware of the importance of self-care, and want it to be more emphasized in the healthcare sector in Japan (see Appendix A for more comments in each theme).

## 4. Discussion

This study aimed to appraise the first-hand experience of healthcare workers in Japan during the COVID-19 pandemic regarding mental health. Our participants reported that they have experienced increased levels of stress and loneliness, which were managed by their limited coping strategies. They also noted that supportive workplace communication and acknowledgement were helpful to their mental health, and an understanding of self-care as an essential factor for the mental health of healthcare workers in Japan. Findings are discussed in turn. 

In line with other countries, healthcare workers in Japan also suffered from heightened stress and loneliness (T1). Uncertainty, increased workload, fear of being infected, and social pressure were noted as key factors causing their mental distress. Indeed, the ability to tolerate uncertainty was highlighted as a preventative construct against burnout, among healthcare workers in COVID [41]. An established and informed protocol was noted as a positive wellbeing factor among Japanese nurses, reducing uncertainty [21]. The Job Demand-Control-Support model that posits the job characteristics influence employee wellbeing may help explain our findings [8]. During COVID, healthcare workers have faced an increased workload (while recognizing some healthcare workers, especially those not treating COVID patients, had reduced workload), restricted work environment, fear of being infected (as the study was before the implementation of vaccine), social pressure (e.g., stigma), and uncertainty (Demand). However, those who gained information and clarity about how they should operate (Control) and receive support from the organization (e.g., frequent update) (Support) maintained a relatively high level of wellbeing. What is novel in our study, though, may be that in the COVID-19 pandemic, what healthcare workers can control was limited. This may help explain why communication and acknowledgement (T3) and self-care (T4) were noted as a positive mental health resource by the participants, compensating for the lack of the control factor. During the early phase of the pandemic (between the second and third waves), healthcare workers experienced stress and loneliness, and the interventions to enhance a sense of control and support may be particularly helpful to reduce those negative mental health outcomes. 

Healthcare workers were not able to use their usual coping strategies during the pandemic (T2), which exacerbated their mental health. Workplace chitchats were reduced due to the workspace separation. Indeed, although office chitchats could hinder employees’ concentration, these informal conversations were helpful to mental health [42]. Social gatherings including having a drink with colleagues, which many participants reported as important to build good colleague relationships, were not allowed, damaging the workplace relationships. High-quality workplace relationships were associated with better staff wellbeing and lower stress among Vietnamese nurses [43]. Likewise, a sense of teamwork was identified as a protective factor for mental health under enhanced work pressure [44]. Evolutionary Theory claims that these informal conversations or chitchats play a key role in facilitating cooperation in the group [45]. Our participants reported that those common means to maintain workplace relationships (e.g., office chitchats, social gatherings) were prohibited during COVID, which sabotaged their meaning in work, namely hatarakigai, and challenged their coping skills with workplace stress.

In this difficult situation, where stress and loneliness were increased (T1) and the coping was limited (T2), healthcare workers recognized communication and acknowledgement as helpful mental health resources (T3). A workplace atmosphere in which employees feel able to discuss concerns, i.e., psychological safety, is essential for staff wellbeing and effective patient care [46]. Moreover, talking with other healthcare workers who are in a similar situation to them relates to common humanity, one of the three components of self-compassion [47], reducing stress. Similarly, participants reported that a sense of being acknowledged by colleagues, patients and community was conducive to their mental health. Self-Determination Theory holds that intrinsic motivation is associated with higher wellbeing, whereas extrinsic motivation is associated with lower wellbeing [48]. Consistent with a previous systematic review [49], our sample of healthcare workers in Japan found recognition and acknowledgement, both categorized as intrinsic rewards, are helpful to their mental health. As healthcare workers in general have higher intrinsic motivation than other occupations [50], they may find intrinsic rewards even more fulfilling. Relatedly, it is noteworthy that the Japanese government offered monetary compensation (extrinsic reward) for all employees in healthcare, which was negatively regarded by healthcare workers (e.g., underestimating their professionalism) whereas positively regarded by administrative staff (e.g., feeling appreciated and recognized) in our sample. These findings suggest that organizations need to be aware of employees’ intrinsic rewards and offer them appropriately. As many participants noted that their professional identity and meaning in work were important, future research should investigate interventions to enhance those positive constructs. 

Lastly, a sector-level understanding of self-care was emphasized as a possible solution for their challenging mental health (T4). As noted above, the control domain in the Job Demand-Control-Support model was compromised during the pandemic, leading to reliance on support. In addition to the workplace support (T3), self-care to support oneself is essential for healthcare workers to maintain wellbeing and good patient care [15]. However, participants reported difficulties implementing their self-care strategies in their workplace, often related to Japanese work culture. This accords to a recent study that identified that Japanese employees’ long working hours were associated with the organizational factors such as team norms and leadership [51], indicating that an organization’s or manager’s understanding of self-care needs to be established for each healthcare worker to care for themselves. While the national policies to stop long working hours have been implemented (e.g., Karoushi Prevention in 2014, Work Style Reform in 2020), traditional cultural value still favors long working: long overtime hours were positively associated with work vigor among Japanese male workers [52]. Indeed, working hours alone do not represent self-care; however, these positive regards on long working can support the participants’ comments about difficulties implementing self-care. Culture and self-care need to be further evaluated [53] to identify a better approach to embed self-care into the Japanese healthcare sector. Moreover, recognizing guilt and shame associated with self-care in other countries [15], our findings highlighting the importance of a sector-level and organizational understanding of self-care may not be limited to Japan (e.g., [54,55]). Further investigation is needed. 

While this research offers helpful insights, limitations should be noted. First, our sample relied on doctors and relatively young professionals, therefore the representativeness of healthcare workers was not high. More diverse samples are needed (e.g., inclusion of other healthcare workers). Nonetheless, our findings will be of interest to healthcare workers in general, appraising their mental health status in the COVID-19 pandemic. Second, the recruitment was done through self-selection: those who were not interested in nor concerned with mental health might not have participated. Third, the analysis was carried out by one author, and all co-authors are healthcare workers: bias might have been present. Fourth, the pandemic is still ongoing, therefore their mental health after the interviews was not considered in this study. 

## 5. Conclusions

Mental health of healthcare workers was negatively impacted by the COVID-19 pandemic. This qualitative study appraised the first-hand experience of healthcare workers in Japan regarding the mental health and coping strategies. Our participants reported that the levels of stress and loneliness were increased, while their coping strategies were limited. Intrinsic rewards such as workplace communication and acknowledgement of their work were identified as positive resources for their mental health. Self-care was highlighted as a possible solution for the challenging mental health in this essential workforce. Our findings can help healthcare organizations and managers identify effective measures to protect employee wellbeing in this crisis.

## Figures and Tables

**Table 1 ijerph-19-00568-t001:** List of participants (*n* = 24).

No *	Age	Sex	Role	Work Setting	Work Experience (Yr)	Weekly Working Hrs	Treat COVID Patients	Regional Level of Infection **
P1	31	M	Doctor	Clinic	7	32.5	Y	Low
P2	29	F	Doctor	Clinic	6	45	Y	Low
P3	35	F	Doctor	Hospital	6	58	Y	Low
P4	34	F	Doctor	Hospital	8	45	Y	Low
P5	32	F	Doctor	Hospital	9	50	N	High
P6	32	M	Doctor	Hospital	9	50	N	High
P7	36	F	Doctor	Hospital	10	55	N	Low
P8	35	M	Doctor	Hospital	12	60	Y	High
P9	49	M	Doctor	Other	16	60	N	High
P10	31	F	Doctor	Hospital	7	80	Y	Low
P11	30	F	Doctor	Clinic	7	55	N	Low
P12	36	F	Doctor	Hospital	8	25	N	Low
P13	34	M	Doctor	Hospital	10	70	Y	High
P14	48	M	Doctor	Clinic	23	60	N	Low
P15	48	F	Nurse	Hospital	28	48	Y	High
P16	43	F	Nurse	Hospital	22	50	Y	High
P17	34	M	Physiotherapist	Hospital	10	40	N	Low
P18	29	M	Physiotherapist	Hospital	6	40	N	Low
P19	27	M	Physiotherapist	Home care station	5	45	N	Low
P20	31	M	Physiotherapist	Home care station	9	40	N	Low
P21	28	N	Physiotherapist	Home care station	6	42	N	Low
P22	27	M	Physiotherapist	Hospital	5	40	N	Low
P23	34	M	Administrator	Clinic	3	40	N	Low
P24	30	M	Administrator	Hospital	5	65	N	Low

* ‘P’ = ‘Participant’. ** Prefectures included in the Second State of Emergency (8 January to 21 March 2021) at the time of the interview were rated as high.

**Table 2 ijerph-19-00568-t002:** Summary of findings.

No	Theme (Corresponding RQ)	Example Participant Excerpt
1	Increased Stress and Loneliness (RQ1)	There were no established guidelines nor legal support and protection for us. I had to look up many things and interpret various pieces of information. … I also had to report our cases to the management. These have significantly increased my workload (P1).
2	Reduced Strategies for Coping (RQ1)	We used to have a lot of chitchats, for example, at the end of our shifts. While writing a daily report, we also talk about how our families are or what we did on a weekend. … During a shift, sometimes we must have direct, negative or intense conversations, but chitchats will help retain our relationship: you know that the person doesn’t dislike you (P22).
3	Communication and Acknowledgement as a Mental Health Resource (RQ2)	It is very helpful to connect with healthcare workers who are in a similar circumstance to me. Now we use video calls to connect with such colleagues, sharing what is happening or giving advice to each other. … This kind of conversation happens organically in the face-to-face context, but now we need a video call to do that (P3).
4	Understanding of Self-Care (RQ3)	My line manager believes that if we don’t take good care of ourselves, we cannot take care of others. He supports self-care, which positively impacts our workplace culture. I am very thankful to him for that (P4).

RQ1: COVID impact on mental health. RQ2: Coping. RQ3: Improve mental health. ‘RQ’ = Research Question. ‘P’ = ‘Participant’.

## Data Availability

The data presented in this study are available on request from thecorresponding author. The data are not publicly available due to ethical restrictions.

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
