# Peer review of "Qualitative Investigation into the Mental Health of Healthcare Workers in Japan during the COVID-19 Pandemic"

_ijerph, 2022, doi:10.3390/ijerph19010568_

Round 1

Reviewer 1 Report

Thank you for the opportunity to review this manuscript. Upon review of the manuscript, there are important methodological issues that preclude major revision.

Author Response

Response Letter

Manuscript ID: ijerph-1477650

"Qualitative investigation into the mental health of healthcare workers in Japan during the COVID-19 pandemic”

Dear Reviewer 1,

Thank you for your helpful feedback. We have systematically revised our manuscript addressing the points you have raised. Please see our responses below. We hope this revised paper is now acceptable for publication. We extend our sincere gratitude to you for your feedback that has significantly helped to strengthen the paper.

Reviewer 1

Reviewer 1’s comment 1

Thank you for the opportunity to review this manuscript. The manuscript addresses a topical issue that would be of interest. Upon review of the manuscript, there are important methodological issues that preclude major revisions and that is where I will stop.

The abstract lacks the methodology used in the work.

Authors’ response 1-1

Thank you for your feedback. In line with your comment, now the methodology is added in the abstract. 

Reviewer 1’s comment 2

It gives an account of a text that addresses key issues in a general and not very specific way, for example communication in the workplace. It is not well defined and is broad.

Authors’ response 1-2

In line with your comment, now these are defined and justified (L.48-54 & 300).

Reviewer 1’s comment 3

The text is untidy in its ideas and in a key part which is the statement of the objectives of the study. The authors describe the purpose of the study with several objectives written in different parts of the text (lines 64 and 96). This dynamic disorients the reader and makes it difficult to understand. It is stated: "One point of focus in our study is to examine the mental health impact of work place communication". Further on it is stated "How Japanese health care workers view and practice self-care during the COVID-19 pandemic has not yet been examined....". Consequently, this study aimed to assess the first-hand experience of mental health among health care workers in Japan". In this regard, I recommend that you consult some articles, some of the author's own:

Unadkat, S., & Farquhar, M. (2020). Doctors' wellbeing: self-care during the covid-19 pandemic. Bmj, 368.

De Maria, M., Ferro, F., Ausili, D., Alvaro, R., De Marinis, M. G., Di Mauro, S., ... & Vellone, E. (2020). Development and psychometric testing of the self-care in COVID-19 (SCOVID) scale, an instrument for measuring self-care in the COVID-19 pandemic. International journal of environmental research and public health, 17(21), 7834.

Kotera, Y. (2021). De-stigmatising self-care: Impact of self-care webinar during COVID19. International Journal of Spa and Wellness, 1-5.

Authors’ response 1-3

Thank you for your helpful suggestions. To enhance coherence and consistency of the manuscript, explanatory statements with justification are now added (L.48-54), and wording of each section is now revised. Indeed, the sentence about workplace communication was confusing, therefore now removed.

Reviewer 1’s comment 4

The authors do not identify the study design, noting the techniques used: semi-structured interviews. The authors should review the purpose and qualitative research method used, or whether this study is a qualitative descriptive study that used a content analysis approach. This issue should be resolved and clarified in the manuscript.

Authors’ response 1-4

In line with your comment, more details of our approach are now added (L.119-123). 

Reviewer 1’s comment 5

What were the criteria for the selection of the sample used for the study, why these professional profiles and not others, to which health area/health service/health territory do the interviewed professionals belong. There are under-represented categories. It is not clear which inclusion and exclusion criteria were used. 

Authors’ response 1-5

In line with your comment, the eligibility criteria of the study are added, as approved in our ethics form (L.121-123).

Reviewer 1’s comment 6

The fact that the data analysis was carried out by a single person is an important limitation in the presentation of the results. The tool used for the coding process should be included, especially since the data analysis was carried out by a single researcher.

Authors’ response 1-6

In line with your comment, now the process is clarified: our use of data corpus and mind-mapping is noted. Moreover, this has been added to the limitations (L.181).

Reviewer 1’s comment 7

With regard to the type of some of the questions used, for example "Did you get clear communication from the management", the participant is directed on how to respond. The questions direct participants to respond in the sense that something has been affected in their experience. Participants need to determine if something has been affected. A better question would be: "Tell me about your experiences with planning/management ..... ". On the other hand, verbatim should not be overused, each verbatim should be carefully selected. 

Authors’ response 1-7

Thank you for your comment. Regarding the wording in our interview question, this cannot be changed at this point as it was approved by the ethics committee and asked to the participants. We will note this in our team research agenda for future work. 

In line with your comment about overuse of quotes, now the number and amount of the quotes are reduced. Thank you.

Reviewer 1’s comment 8

In the discussion section, the results should not be shown again as in this study, nor should the objectives of the study be recalled.

Authors’ response 1-8

In line with your comment, now sentences here are either removed or reduced (L.377-381).

Reviewer 1’s comment 9

It would be necessary to include the approval of the ethics committee mentioned.

Authors’ response 1-9

We agree; it is noted in the methods section (L.132). Thank you.

Reviewer 2 Report

Concerning the article titled: Qualitative investigation into the mental health of healthcare
workers in Japan during the COVID-19 pandemic, I have to mention the following.

Strengths of the article

  • The issue of the covid-19 pandemic and how it affects health professionals is a topic of great interest to the scientific community. Therefore, any research that deals with this issue is of scientific interest.
  • The covid-19 pandemic has forced all health professionals to work a lot of hours and face death on a daily basis. Consequently their mental health has been affected. The article therefore deals with the issue of mental health, which is very interesting.
  • The researchers of the article used a qualitative research method. This research method helps the researcher to understand deeper views, attitudes and behaviors of the sample. (Line 129-135).  
  • Finally, the authors used thematic analysis to analyze the interview data based on Braun and Clarke published work. (Lines 143-149).

  Weaknesses of the article

  • The research has a very small sample. Of course, because it is a qualitative method, the sample is usually small compared to a quantitative one.
  • Also, a problem that this research has is the composition of the sample. Most people are doctors as opposed to nurses (2) and administrators (2). Moreover, the sample consists only of young professionals. The results may have been different if there were other age groups. Third, all co-authors are healthcare employees; therefore bias might have been present.

Author Response

Response Letter

Manuscript ID: ijerph-1477650

"Qualitative investigation into the mental health of healthcare workers in Japan during the COVID-19 pandemic”

Dear Reviewer 2,

Thank you for your helpful presentation of the strengths and weaknesses of our manuscript. Here, we report how we have addressed the weaknesses you had noted. 

Reviewer 2’s comment 1

The research has a very small sample. Of course, because it is a qualitative method, the sample is usually small compared to a quantitative one. Also, a problem that this research has is the composition of the sample. Most people are doctors as opposed to nurses (2) and administrators (2). Moreover, the sample consists only of young professionals. The results may have been different if there were other age groups. 

Authors’ response 2-1

In line with your comment, now it is noted in the limitation section (L.466-473). Thank you for your helpful feedback.

Reviewer 3 Report

This interesting paper investigates healthcare workers’ mental health during the COVID-19 pandemic. This paper is in line with the themes covered by this journal, and it seems that the readers of this journal are highly interested.

However, it seems that some points need to be corrected and/or amended, so please refer to them and correct and/or amend the manuscript.

The authors used the Japanese words “hatarakigai” in the abstract. It should be translated into easy-to-understand English words.

In line 86, it seems unnecessary to start writing at the beginning, “Lastly, …” (“Lastly” is unnecessary).

In line 104, it seems unnecessary to start writing with at the beginning, “Accordingly, …” (“Accordingly, …” is not necessary).

In line 118, the authors described “Range 49-27 years.” I wonder if there is any intention in describing “49-27” instead of “27-49.”

Does “No. 1, No. 2, ...” in Table 1 indicate “P1, P2, ...” in the text, respectively? Is “P” an abbreviation for “Participant”? If so, it should be made easier to understand by changing the description (for example, changing from “1, 2, …” to “P1, P2, …” in Table 1).

The “hospital” should be changed into “Hospital” in the Work Setting of No. 22 in Table 1.

The description from line 110 to line 113 in the “Study Aims” is exactly the same as the subsequent lines 174 to 177. I recommend that authors modify (change) the description from line 110 to line 113 in that part as appropriate.

As mentioned above, try to make it easier to understand what “P1, P21, ...” from line 233 onward indicates.

The specific dialogue of each participant dominates the result. Please consider whether it is possible to summarize these as appropriate and describe them simplified. In addition, please consider whether these dialogues can be posted as supplementary materials.

The Job Demand-Control-Support model is mentioned for the first time on line 389 of the Discussion section—a brief explanation of what kind of model should be added. In addition, it should be described and explained in advance in the introduction section, not in the discussion section, for the first time.

The contents of lines 461–465 (COVID-19 increased the levels … protect the mental health of healthcare workers) are repeatedly described in other parts such as the conclusion section, and I get the impression that there are many repetitions. Please make appropriate corrections.

Author Response

Response Letter

Manuscript ID: ijerph-1477650

"Qualitative investigation into the mental health of healthcare workers in Japan during the COVID-19 pandemic”

Dear Reviewer 3,

Thank you for your helpful feedback. We have systematically revised our manuscript addressing the points you have raised. Please see our responses below. We hope this revised paper is now acceptable for publication. We extend our sincere gratitude to you for your feedback that has significantly helped to strengthen the paper.

Reviewer 3

Reviewer 3’s comment 1

This interesting paper investigates healthcare workers’ mental health during the COVID-19 pandemic. This paper is in line with the themes covered by this journal, and it seems that the readers of this journal are highly interested.

However, it seems that some points need to be corrected and/or amended, so please refer to them and correct and/or amend the manuscript.

The authors used the Japanese words “hatarakigai” in the abstract. It should be translated into easy-to-understand English words.

Authors’ response 3-1

Thank you for your feedback. In line with your comment, now hatarakigai is explained in the abstract.

Reviewer 3’s comment 2

In line 86, it seems unnecessary to start writing at the beginning, “Lastly, …” (“Lastly” is unnecessary).

Authors’ response 3-2

In line with your comment, it is now removed.

Reviewer 3’s comment 3

In line 104, it seems unnecessary to start writing with at the beginning, “Accordingly, …” (“Accordingly, …” is not necessary).

Authors’ response 3-3

In line with your comment, it is now removed. Thank you.

Reviewer 3’s comment 4

In line 118, the authors described “Range 49-27 years.” I wonder if there is any intention in describing “49-27” instead of “27-49.”

Authors’ response 3-4

Thank you for spotting this. It should be 27-49. It is now amended.

Reviewer 3’s comment 5

Does “No. 1, No. 2, ...” in Table 1 indicate “P1, P2, ...” in the text, respectively? Is “P” an abbreviation for “Participant”? If so, it should be made easier to understand by changing the description (for example, changing from “1, 2, …” to “P1, P2, …” in Table 1).

Authors’ response 3-5

Thank you for your helpful suggestion. Now Table 1 is revised, noting ‘P1’, ‘P2’... noting that ‘P’ indicates ‘Participant’.

Reviewer 3’s comment 6

The “hospital” should be changed into “Hospital” in the Work Setting of No. 22 in Table 1.

Authors’ response 3-6

Thank you for spotting this. Amended.

Reviewer 3’s comment 7

The description from line 110 to line 113 in the “Study Aims” is exactly the same as the subsequent lines 174 to 177. I recommend that authors modify (change) the description from line 110 to line 113 in that part as appropriate.

Authors’ response 3-7

In line with your comment, now the description at L110-113 is amended.

Reviewer 3’s comment 8

As mentioned above, try to make it easier to understand what “P1, P21, ...” from line 233 onward indicates.

Authors’ response 3-8

In line with your comment, clarification is now added to Table 2 and L.245.

Reviewer 3’s comment 9

The specific dialogue of each participant dominates the result. Please consider whether it is possible to summarize these as appropriate and describe them simplified. In addition, please consider whether these dialogues can be posted as supplementary materials.

Authors’ response 3-9

Thank you again for your helpful suggestions. Now the amount of quotes is reduced, and where necessary, additional description is added. Supplementary file is now created.

Reviewer 3’s comment 10

The Job Demand-Control-Support model is mentioned for the first time on line 389 of the Discussion section—a brief explanation of what kind of model should be added. In addition, it should be described and explained in advance in the introduction section, not in the discussion section, for the first time.

Authors’ response 3-10

In line with your comment, this model is now introduced in the introduction section with a brief explanation.

Reviewer 3’s comment 11

The contents of lines 461–465 (COVID-19 increased the levels … protect the mental health of healthcare workers) are repeatedly described in other parts such as the conclusion section, and I get the impression that there are many repetitions. Please make appropriate corrections.

Authors’ response 3-11

In line with your comment, this part is now removed, and other parts that are repetitive are also revised.

Round 2

Reviewer 3 Report

The authors carefully examined the matters I pointed out last time (previous reviewer comments), and I confirm that the necessary corrections were made. I thought this article is a good treatise for publication. Thank you for responding appropriately to my requests.

This interesting paper investigates healthcare workers’ mental health during the COVID-19 pandemic. This paper is in line with the themes covered by this journal, and it seems that the readers of this journal are highly interested.